# Relationship of Cycling Power and Non-Linear Heart Rate Variability from Everyday Workout Data: Potential for Intensity Zone Estimation and Monitoring

**DOI:** 10.3390/s24144468

**Published:** 2024-07-10

**Authors:** Stefano Andriolo, Markus Rummel, Thomas Gronwald

**Affiliations:** 1AI Endurance Inc., Hamilton, ON L8P 0A1, Canada; 2Institute of Interdisciplinary Exercise Science and Sports Medicine, MSH Medical School Hamburg, 20457 Hamburg, Germany; 3G-Lab, Faculty of Applied Sport Sciences and Personality, BSP Business and Law School, 12247 Berlin, Germany

**Keywords:** interbeat intervals, RR-intervals, HRV, DFA, monitoring

## Abstract

The short-term scaling exponent alpha1 of detrended fluctuation analysis (DFA-a1) of heart rate variability (HRV) has been shown to be a sensitive marker for assessing global organismic demands. The wide dynamic range within the exercise intensity spectrum and the relationship to established physiologic threshold boundaries potentially allow in-field use and also open opportunities to provide real-time feedback. The present study expands the idea of using everyday workout data from the AI Endurance app to obtain the relationship between cycling power and DFA-a1. Collected data were imported between September 2021 and August 2023 with an initial pool of 3123 workouts across 21 male users. The aim of this analysis was to further apply a new method of implementing workout group data considering representative values of DFA-a1 segmentation compared to single workout data and including all data points to enhance the validity of the internal-to-external load relationship. The present data demonstrate a universal relationship between cycling power and DFA-a1 from everyday workout data that potentially allows accessible and regular tracking of intensity zone demarcation information. The analysis highlights the superior efficacy of the representative-based approach of included data in most cases. Validation data of the performance level and the up-to-date relationship are still pending.

## 1. Introduction

Knowledge of an athlete’s individual exercise intensity zones is crucial for tailoring effective endurance training plans. To determine physiologic thresholds for exercise intensity distribution, various methods have been employed based on different physiological measures, such as gas exchange data, blood lactate concentration, and heart rate (HR) variability (HRV) kinetics [1,2,3,4,5,6,7]. The first physiological boundary transition has been described as the aerobic threshold (AeT) and the second boundary has been referred to as the anaerobic threshold (AnT) to provide the demarcation in a classic three-zone model (e.g., zone 1 as “moderate” intensity: ≤AeT, zone 2 as “heavy” intensity: >AeT to ≤AnT, zone 3 as “severe” intensity: >AnT). Regularly updating these transitions is essential to avoid training with the wrong intensity targets. The disadvantages of exceeding intensity targets may include glycogen depletion [3], prolonged cardiac parasympathetic recovery [8,9], and gastrointestinal barrier disruption [10] along with the potential of overall central and local muscular fatigue [11,12]. However, information about these physiological transitions typically requires laboratory testing and/or specific exercise protocols.

In that regard, it could be shown that the short-term scaling exponent alpha1 of detrended fluctuation analysis (DFA-a1) [13,14,15,16] as a non-linear metric of HRV may be a sensitive marker for assessing global organismic demands during acute endurance exercise with a wide dynamic range, encompassing the moderate, heavy, and severe exercise intensity domains [17,18,19,20,21,22,23,24]. DFA-a1 quantifies the fractal scale as a qualitative pattern of ANS regulation and represents correlation properties of HR time series in cardiac beat-to-beat intervals with the maintenance of basic stability of the control systems between order (persistence) and disorder (change) in the context of homeodynamics during resting conditions with values around 1.0 [25,26,27,28,29,30]. During exercise, DFA-a1 depicts strongly correlated patterns (values well above 1.0, periodic behavior) at low-intensity exercise, transitions to fractal patterns (value at around and below 1.0) at moderate exercise intensities, and drops to uncorrelated and anticorrelated patterns at the highest intensities in the severe domain (values around and below 0.5, loss of fractal dynamic toward random and anticorrelated behavior). Importantly, this method does not require a maximal exercise test, and data can be derived from a chest belt with a sensor device (e.g., H10 of Polar Electro GmbH (Büttelborn, Germany), Movesense Medical of Movesense Ltd. (Vantaa, Finland)), which, therefore, allows in-field use and also opens opportunities to provide real-time feedback on exercise intensity. Given these properties, a proposal was made based on the signal-theoretical background to utilize this metric as a biomarker for exercise intensity distribution, including discrete numerical values (AeT: DFA-a1 of 0.75, AnT: DFA-a1 of 0.5) that correspond to established physiologic threshold boundaries [24].

The present report tries to expand this idea by using everyday workout data to obtain up-to-date intensity zone demarcation information with the relationship between cycling power and DFA-a1. The lower the DFA-a1, the higher the organismic demands during exercise, and as a measure of global internal load, DFA-a1 is expected to correlate strongly with the external load metric power at comparable internal and external conditions for a given acute and chronic physiological status of the athlete. The aim of this analysis was to further apply a new method of implementing workout group data considering representative values of DFA-a1 segmentation compared to single workout data and including all data points to enhance the validity of the internal-to-external load relationship.

## 2. Materials and Methods

### 2.1. Participants

The analysis described here does not rely on laboratory data but instead daily workout data of cycling athletes collected through the AI Endurance platform app between September 2021 and August 2023. An initial pool of 3123 workouts across 21 users was included. All 21 users are male with an average age of 52.5 years (SD ± 8.5, range: 34–66). For reference, the average number of workouts for each user is 148.7 (SD ± 129.1, range: 28–380), spanning an average of 312.3 days (SD ± 198.7, range: 78–689) and yielding an average of 0.47 workouts per day for each user (SD ± 0.25, range: 0.13–0.91). Written informed consent about data availability and processing was obtained from each participant. All participants have agreed to share their data in AI Endurance for research purposes. All participant’s data were anonymized. Due to the applied standards, the local ethics board of MSH Medical School Hamburg waived the requirement for ethical approval for the current study according to the guidelines for ensuring good scientific practice of MSH Medical School Hamburg. Ethical approval for the used approach of heart rate variability analysis was obtained through the local ethics committee of MSH Medical School Hamburg (reference No. MSH-2022/172). This study was carried out in accordance with the principles set forth in the most recent revisions to the Declaration of Helsinki.

### 2.2. Study Design

The data collection was entirely passive as the participants did not follow a particular set of study instructions during their workouts. Focusing on cycling power and utilizing a diverse range of workout data (including ERG mode, a smart trainer mode that automatically controls the resistance such that the athlete has to exercise at a pre-determined power) collected across varying intensities and conditions, a new method to elucidate the correlation between power and DFA-a1 was devised. The new method is used to define representatives of each DFA-a1 to power a data cluster via a simple averaging procedure that alleviates the effects of cardiac lag that otherwise creates noise in the DFA-a1 to power relationship. This method has been applied to determine the correlation between using data from single workouts first and then data on workout groups. The latter has been obtained by merging data of all workouts executed within a window of 10 days. Only groups containing at least 4 workouts have been selected.

### 2.3. Data Analysis

Timestamped cycling power and RR interval data were imported into the AI Endurance platform via Garmin or Suunto recording devices. The power and RR interval data were recorded with unknown devices. Data were only used if the RR interval data were of sufficient quality (see section “data quality”). In AI Endurance, RR interval data were controlled and corrected for artifacts (long, short, missed, extra, and ectopic beats) [31,32,33] and afterward detrended [34] with the “smoothness priors” of Lambda = 500. DFA-a1 was calculated with “short term fluctuations” if 4 ≤ n ≤ 16 beats. For the DFA-a1 calculation, the window width was set to 120 s, with a recalculation grid interval performed every 5 s. The same window width of 120 s was applied to calculate the cycling power (in Watts) as the mean over this window, every 5 s. To analyze the correlation between power and the internal load marker DFA-a1, certain prerequisites must be met for the data under scrutiny.

Data Quality: Artifacts—e.g., missed beats by HR monitors—can significantly affect the DFA-a1 calculation. Hence, the focus of the data analysis was on including workouts with a maximum of 5% artifacts. From the initial data pool, this criterion narrows down the dataset to 2096 workouts (still 21 users), forming 411 10-day workout groups.Fatigue Consideration: To mitigate the impact of fatigue, only minutes 5 to 20 of each workout were included in the data analysis. The first 5 min of data were excluded as they are more prone to HRV artifacts due to the HR monitor potentially not having built up sufficient moisture from the athlete’s sweat and the potentially suboptimal position of the chest strap that gets adjusted during the first few minutes of exercising.Data Consistency: Inconsistent data, especially during periods without pedaling (where power equals 0), can skew the results. Indeed, while stopping spinning for a few seconds occasionally is not a problem, it becomes one when it is performed for longer times and/or very frequently. In such a situation, a lot of data points with the same power value of 0 and different values of DFA-a1 would be included, significantly spoiling the correlation. To maintain consistency, workouts with prolonged or frequent periods of no pedaling (exclusion of datasets pedaling < 90% of the dataset) were excluded, and data points with a power of 0 were discarded.Data Range and Intensity: For a meaningful correlation between power and DFA-a1, a sufficient range of the internal load marker is necessary. Data inclusion focuses on datasets where at least 50% of the data points were within the dynamic range (DFA-a1 < 1.0), which indicates a significant effort level. Data with mostly DFA-a1 > 1.0, typically corresponding to very easy rides, breaks, or resting conditions in healthy individuals, were not processed [22].

The last criterion is the most restrictive, reducing the number of eligible single workout datasets to 554 across 17 users and workout groups to 73 across 11 users. For reference, the average number of workouts for each user is now 32.6 (SD ± 55.3, range: 1–225), while the list of workout groups for each user is 1, 36, 5, 14, 1, 2, 6, 3, 1, 3, and 1, yielding an average number of groups per user of 6.6 (SD ± 10.5). The reduction in the number of workout groups is also influenced by the decision to exclude groups with fewer than four workouts to minimize the influence of individual workouts on the overall group analysis.

Even after meticulous data selection, directly correlating power and DFA-a1 using all data points available typically yields weak correlations (for a definition of correlation coefficients, see Section 2.4), as illustrated by the blue histograms in Figure 1. This is because while data points are a list of (P, DFA-a1) pairs collected at the same time, there is a physiological lag in the cardiac response to an increase/decrease in power output as the external load measure. (Determining the optimal lag between power time series and DFA-a1 series through cross-correlation presented methodological challenges, leading to ambiguous criteria to pinpoint the “best correlation” and its corresponding lag value. Consequently, specific lag determination methods are not detailed in this study.) This physiological lag means that when an athlete suddenly increases power, there are data points with high power but still high DFA-a1 values, as the latter takes time to catch up. Similarly, during a decrease in power (especially after intense effort), there are points with low power but still low DFA-a1. These inconsistencies lead to poor correlations. An example would be an athlete attempting a short power burst (severe domain) after riding at low exercise intensity (moderate domain) and going back to riding at low intensity (moderate domain) after this effort. The raw (P, DFA-a1) data of this example exhibits data points outside the expected negative correlation.

A sound approach to the correlation analysis might be used to concentrate on periods where DFA-a1 values demonstrate stability—defined as intervals of 90–120 s with minimal variance in consecutive DFA-a1 readings—and to analyze corresponding power series within these intervals. While theoretically effective under controlled conditions, the practical application of this methodology is significantly limited by the unstructured nature of the data. Among 2096 analyzed workouts, only 101 exhibited the specified stable intervals, with a mere 17 containing more than 1 stable period. This scarcity of suitable data points renders comprehensive statistical analysis unfeasible. The method proposed in this study offers a simpler solution to the aforementioned challenge, preserving the volume of total workout data and effectively uncovering the correlation between power and DFA-a1. This new method (“representative method”) was developed to smooth out problematic data points for the comparison using all data points (“standard approach”).

Data Segmentation: In this study, data points (depicted as blue and light blue data points in Figure 2 and Figure 3) were segmented into intervals based on DFA-a1 levels, delineated by yellow (and purple) dashed lines in these figures. The segmentation methodology varied depending on the dataset. For individual workouts, due to a high sparseness of data points outside the dynamic range (DFA-a1 > 1.0), the data were divided into eight equal-length intervals within the range [a1_min_, a1*], where a1_min_ represents the minimum DFA-a1 value and a1* is the lesser of DFA-a1 = 1.2 and the maximum DFA-a1 value. For workout groups, two distinct regions were defined: within the dynamic range [a1_min_, 1.0], data points were segmented into nine equal-length intervals, and in the range [1.0, a1*] (a1* being the lesser of DFA-a1 = 1.8 and the maximum DFA-a1 value), data were divided into five equal-length intervals.Representative Points: Within each interval, a representative point was established by averaging both power and DFA-a1 (P_avg_, DFA-a1_avg_) for that interval. These representative points are indicated as red data points in Figure 2 and as red and purple data points in Figure 3. For individual workouts, a representative point is calculated only if the interval contains at least eight data points, while for workout groups, the minimum threshold was ten data points. This method enhances the reliability of the representative points, excluding those derived from an insufficient data quantity.Correlation of Representatives: Instead of correlating all individual data points, these representative points were correlated. Initially, correlations were computed for individual workouts using the representatives in Figure 2 (indicated as the last correlation value in each plot title). Subsequently, correlations for workout groups were calculated. In this case, two correlation coefficients were derived for each workout group: one using representatives within the dynamic range (red data points) and another incorporating all representatives, if feasible (including both red and purple data points). These two sets of correlation coefficients are presented as the final two values in the plot titles in Figure 3.

### 2.4. Statistics

Descriptive and analytical statistical analyses were performed for the tested variables using Python (version 3.8). Given the fact that the variables are not normally distributed, and a non-linear relationship is suspected, Spearman’s r correlation coefficient was used to study the relationship between power and DFA-a1. The size of Spearman’s r correlations was evaluated as follows: 0.3 ≤ |r| < 0.5 low, 0.6 ≤ |r| < 0.8 moderate, and |r| ≥ 0.8 high [35]. Additionally, the workout groups were analyzed with bi-parametric linear and hyperbolic regression fits, relating power to DFA-a1, respectively, as P = m × DFA-a1 + q (with the parameters slope m and the intercept q) and the P = s/DFA-a1 + t model (with parameters s and t). Their equations with the actual values of m, q, s, and t are displayed in Figure 3. The goodness of these fits is calculated by the coefficient of determination (R^2^), with a value closer to 1 indicating a more accurate fit.

## 3. Results

The use of the representative method to calculate correlations significantly alters the results when compared to the standard approach. This effect is clearly visible in the histograms in Figure 1, which illustrate an exaggeration of the standard correlation values, leaning towards −1.0. Regarding single workouts (first histogram), the mean for the standard correlation (blue) is −0.25 (SD ± 0.26; range: −0.95–0.41). In contrast, correlations of representatives (red) show a mean of −0.44 (SD ± 0.55; range: −1–1). Notably, while only 3% of workouts have a standard correlation higher than −0.7, this proportion increases to 44% when using representatives. Furthermore, in 74% of cases, the correlation calculated with representatives is stronger than the standard correlation. Delving into specific examples, as depicted in Figure 2, the analysis observes that in common workout scenarios, the correlation derived using the new method is generally stronger than that calculated using all data points. This is particularly evident in the first (A) and third (C) workouts shown. Notably, the third workout exemplifies the method’s efficacy, even in complex exercise scenarios like those performed in ERG mode. The same pattern is observed for workout groups (Figure 1B). The mean values for standard (blue) and representative (red) correlations are −0.32 (SD ± 0.19) and −0.75 (SD ± 0.27), respectively. In this scenario, a significant 66% of workout groups exhibit a correlation higher than −0.7 (red) compared to only 4% in the standard approach. Moreover, in 96% of workout groups, this correlation is stronger. This implies that all users have at least 92% of their workout groups, showing an improvement in correlation calculation with the new method over the standard approach. When considering two different parameter fits (linear and hyperbolic regression) to determine the relationship between power output and physiological response, the analysis does not distinctly prefer either the linear model (P = m × DFA-a1 + q) or the hyperbolic model (P = s/DFA-a1 + t). Both approaches seem equally valid in the context of both the dynamic and the entire DFA-a1 range, as illustrated in Figure 3 (specific examples) and Figure 4 (statistical analysis on fits). In the former case, data show means of 0.59 and 0.64 (SD ± 0.29, SD ± 0.28) for the linear and hyperbolic law, respectively, while in the latter case, data show means of 0.49 and 0.54 (SD ± 0.30, SD ± 0.26), respectively.

## 4. Discussion

The analysis yielded a significant observation: in 74% of the individual workouts examined, the correlation derived from the representative method surpassed that of the standard approach, which involves using the complete set of (P, DFA-a1) data without representative points. When analyzing grouped workouts, this figure escalates to 96%. Moreover, with this new method, 44% of single workouts and 66% of workout groups show a correlation stronger than −0.7 compared, respectively, to the 3% and 4% obtained when all data points are considered (see Figure 1 for detailed statistics). This result highlights the superior efficacy of the representative-based approach in most cases, confirms the expected correlation between power and DFA-a1 (both within single workouts and workout groups, see Figure 1, Figure 2 and Figure 3), and additionally demonstrates a number of advantages in aggregating workouts into groups.

Reduced Impact of Workout Types: Merging different types of workouts, as seen in the second group (B) in Figure 3, minimizes the influence of workout variety. This results in more generalizable insights.Enhanced Significance of Data Points: In the workout groups, data points are more densely packed, making the representatives much more meaningful than in isolated workouts. For example, representatives outside the dynamic range display a clearer behavior, aligning with the expectation that lower power corresponds with lower effort. This phenomenon is evident in Figure 3’s first workout group (A), where workouts with weaker individual correlations (as B in Figure 2) contribute to groups with significantly stronger correlations.Identification of Potential Non-Responders: The approach facilitates the detection of non-responders—specified as athletes whose workouts consistently show very weak correlations across almost all their workout groups. In the present analysis of 11 athletes, data revealed no non-responders.Modeling: Collecting data over several days allows for a meaningful investigation of the short-term relationship between power and DFA-a1. As demonstrated in Figure 3, it was tested with two types of two-parameter functions for each workout group, linear regression and hyperbolic fit, since a single parameter hyperbolic fit did not fit the data sufficiently well. As detailed in Figure 4, neither model showed a remarkable preference, leading to an analysis of the simpler linear model for its straightforwardness: P = m × DFA-a1 + q.

In Figure 3, values of the descriptive parameters of the regression fit m and q are shown for specific examples. These two parameters are not only user dependent but also vary with the level of fitness. For example, if an athlete achieves higher power for a given DFA-a1 over time, i.e., increasing fitness, the parameters m and q will change. This law can be used to track the fitness level of each user across time, and it offers a method to estimate the most current values of power at specific values of DFA-a1 (e.g., P = m × 0.75 + q and P = m × 0.5 + q) without the need for any specific testing protocols but simply out of the most recent workout history. This is a major implication of the present analysis and reveals the potential of tracking an athlete’s exercise intensity zone transitions without a laboratory visit or a testing protocol, only requiring them to enter the dynamic range of DFA-a1 < 1.0 somewhat regularly during their workouts with clean HRV data. When considering the two different parameter fits (linear and hyperbolic regression) to determine the relationship between power and DFA-a1, the analysis does not distinctly prefer either the linear or the hyperbolic model. Given the lack of a clear advantage for one model over the other and the current size of the dataset, this leads towards adopting the linear model for its simplicity and ease of interpretation.

Regarding application scenarios, the DFA-a1 metric has been shown to have utility as a marker of fatigue in recent studies [24,36,37]. For example, DFA-a1 was significantly lower after a 6 h simulated ultramarathon when running at a standardized submaximal speed close to the first ventilatory threshold on a treadmill [37] and was also markedly suppressed at the first ventilatory threshold when assessed during a second incremental test on a treadmill after a short recovery period [38]. In that regard, DFA-a1 may also be useful to inform on the physiological status of an athlete as a surrogate of daily directed training or “training readiness”. With more fatigue, the usual relationship between the correlation properties of HR time series and exercise intensity (here: cycling power) could be shifted, with less correlated patterns (lower DFA-a1 values, indicating autonomic fatigue) occurring at a given external load situation, potentially providing further opportunities for training guidance. Sample case data, data from a reliability study, and data from a pilot study with decreased DFA-a1 values during a standardized warm-up; for example, prior to a planned training session led to the hypotheses that this metric could be used for decision support to modify the training load (intensity and/or volume) of the upcoming session, taking into account contextual data, such as psychometrics [24,39,40]. Potentially, the altered relationship of the external and internal load of daily workout data could provide valuable information about individual performance enhancement without specific testing scenarios in the laboratory and field conditions. Examples are given in Figure 5 for two of the included participants. These plots illustrate the temporal evolution of three distinct power values, corresponding to DFA-a1 values of 1.0, 0.75, and 0.5 within each workout group. This evolution is contextualized against two key metrics: the External Stress Score (ESS) and the power-to-HR ratio. In determining the relationship between power values and DFA-a1 to be used to compute P(DFA-a1 = 1.0), P(DFA-a1 = 0.75), P(DFA-a1 = 0.5), whether linear or hyperbolic, a “best fit” approach is employed. This method selects the model with the coefficient of determination R^2^ nearest to 1.0. However, to ensure reliability, the selected model is only utilized if the coefficient of determination exceeds 0.75, indicating a robust fit.

The ESS serves as a dimensionless indicator of workout load and is defined as the sum over sec_step/3600 × i_step^2^ × 100, where sec_step divides the workout into steps of up to 10 min. Within each step, i_step is defined as the rolling fourth power-weighted average normalized by its threshold setting (defined as a proxy of AnT), i.e., (mean(moving_av(power)^4^))^0.25^/thresh, where moving_av(power) is a 30 s rolling average. Thresh is set either by the user manually in the AI Endurance app or automatically calculated by the user’s historical intensity distribution. For example, one hour at thresh results in an ESS of 100. Unfortunately, we cannot guarantee that the user did not change the thresh setting between workouts used for this study. The reason for dividing the workout into steps of 10 min duration is the inherent non-linearity in time of the fourth power-weighted averaging procedure. Dividing into steps restores linearity and leads to sensible linear behavior at the time of the stress metric, the ESS. For the purpose of maintaining uniformity with data provided by workout groups, ESS calculations are based on a 10-day rolling average. This standardized approach allows for more accurate comparisons and analyses across varying workout regimens. The power-to-HR ratio is defined as the ratio resulting from dividing the average power by the average HR. This ratio is specifically calculated for the time interval extending from the 5th to the 20th minute of each workout. To ensure consistency in our analysis, especially when comparing across different workout groups, this power-to-HR ratio is also evaluated on a 10-day rolling baseline. Where adequate data exist, as evidenced in the historical plots for two users in Figure 5, a notable correlation emerges between the trend of the calculated P(DFA-a1) values and the trend of the ESS. This observation suggests the potential effectiveness of P(DFA-a1) values in monitoring fitness alterations over time. Although insightful, Figure 5 also highlights the essential need for additional data that adhere to all established quality thresholds. Despite the breadth of the current dataset, only a limited portion satisfies the rigorous quality standards necessary for a robust study on fitness tracking. This situation emphasizes the importance of not only collecting a larger volume of data but also ensuring that a larger fraction of data consistently meets the high-quality benchmarks required for a comprehensive and reliable analysis.

## 5. Limitations

Regarding data inclusion, only male users were included, and there was a lack of control over the data recording devices (HR monitors and the cycling power meter) inherent to this kind of study. While RR interval data quality can be largely assessed via artifact inspection, power meter data could have various issues, including the potential switching of power meters between sessions (for example, for an athlete with multiple bikes or smart trainers) or calibration offsets. It is important to note that the developed approach of the representative method does also sometimes reveal only a weak correlation, possibly due to factors like limited data or a focus on certain representative regions (e.g., where DFA-a1 > 1), as illustrated in the last workout (C) in Figure 2. Such weak correlations can also occur in data-richer workouts, as seen in the second workout (B) in Figure 2. If weak correlations are consistently observed across an athlete’s workouts, it might indicate a non-responder pattern. These observations suggest that broader datasets could provide more comprehensive insights. This expectation is indeed confirmed when considering bigger datasets obtained by merging multiple workouts together. In a past analysis [38], it was found that potential “non-responders” for the overall approach also exist. In these cases, an inappropriate suppression of the correlation properties losing the dynamic range of DFA-a1 despite good ECG waveforms and little artifact containment in HR time series could be observed. Here, DFA-a1 values were already markedly suppressed at exercise with moderate intensity during running. The reason for this behavior is unclear, but possible causes include oscillation of the cardiac axis (and, therefore, the precision of the RR interval measurement) due to foot strike-related impact. In the present analysis, a non-responder was defined as an athlete with a poor correlation between DFA-a1 and power. While no non-responder could be identified in this study, some athletes anecdotally already reported markedly suppressed DFA-a1 < 0.5 at the commencement of exercising. Another aspect is to consider the chosen datasets, which are obtained by grouping workouts executed within a short time frame of 10 days. Each workout group is nothing but a 10-day snapshot of data regarding external (power) versus internal (DFA-a1) load and offers a more nuanced understanding of the individual level of performance. However, it is important to limit the time frame, like 10 days, to avoid the influence of physiological adaptations on the short-term analysis.

## 6. Conclusions

The present data demonstrate a universal relationship between cycling power and DFA-a1 from everyday workout data that potentially allows accessible and regular tracking of intensity zone demarcation metrics based on the correlation properties of HR time series without the need for an exercise lab or even a dedicated testing protocol. The analysis highlights the superior efficacy of the representative-based approach of included data in most cases and confirms the expected correlation between power and DFA-a1. A bigger dataset may reveal the most suitable model to represent the universal relationship between power and DFA-a1 in the future. Validation data regarding the change in physical performance and the existing relationship between the included internal and external load data are still pending.

## Figures and Tables

**Figure 1 sensors-24-04468-f001:**
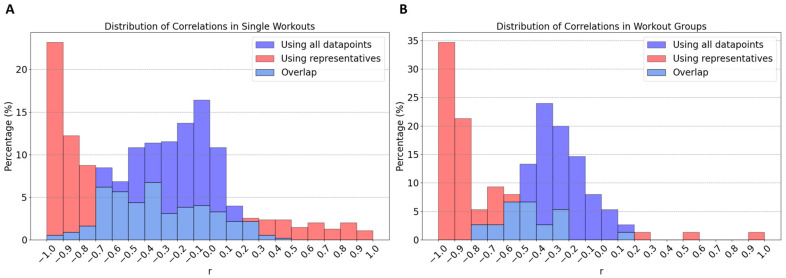
Distribution of correlation coefficients (r) obtained in all single workouts (**A**) and workout groups (**B**). Blue shows the correlations with the standard approach obtained using all data points, red shows the correlation obtained with the representative method. Since two correlations of representatives were computed for each workout group, the red histogram in (**B**) refers to the strongest correlation between these two values.

**Figure 2 sensors-24-04468-f002:**
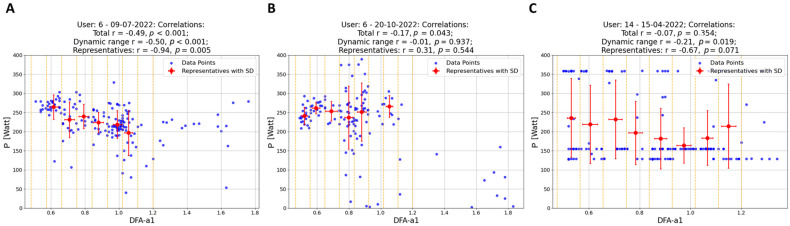
Correlations (and respective *p*-values) between power (in Watts) and DFA-a1 within single workouts. The first 2 workouts (**A**,**B**) belong to the same participant, and the third (**C**) corresponds to a workout executed in ERG mode from another participant. Blue points are the (P, DFA-a1) data collected and are used to compute the correlation of all data points (total) and the correlation of the dynamic range (considering data points with DFA-a1 < 1.0). Red points are the representatives of each DFA-a1 (and standard deviation, SD) interval (marked by yellow dashed lines, up to DFA-a1 = 1.2) and are used to compute the correlation following the newly implemented method.

**Figure 3 sensors-24-04468-f003:**
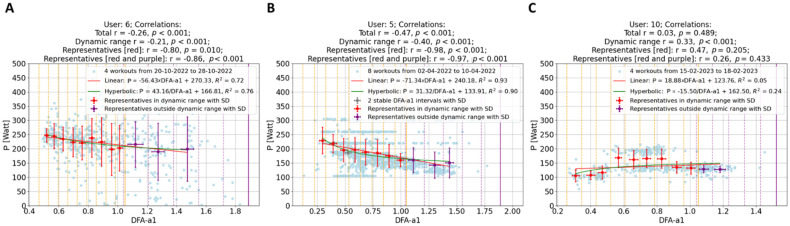
Correlation of (P, DFA-a1) data points obtained by merging workouts into groups. The groups shown correspond to different users and contain 4 (**A**), 8 (**B**), and 4 (**C**) workouts, respectively. Light blue points represent all workout data collectively, while red and purple points are the representatives (and standard deviation, SD) computed inside and outside the dynamic range, respectively. These are used to obtain the “red” and “red + purple” correlation values. Notice that the first group contains the second workout shown in Figure 2, and the second group refers to an athlete often using ERG mode. Image C presents the least successful result, the only group showing a positive correlation, which belongs to a user who has a total of 13 workout groups. Additionally, each group is analyzed with linear and hyperbolic fits, and their corresponding equations are displayed. Notice that power is presented as “P” in the formulas.

**Figure 4 sensors-24-04468-f004:**
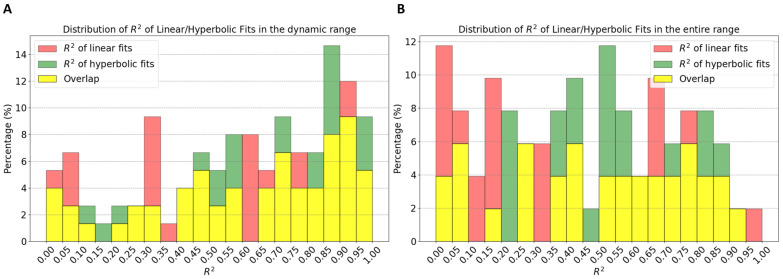
Distribution of the coefficient of determination (R^2^) of the linear and hyperbolic fits in the dynamic range (**A**) and the entire region for DFA-a1 (**B**).

**Figure 5 sensors-24-04468-f005:**
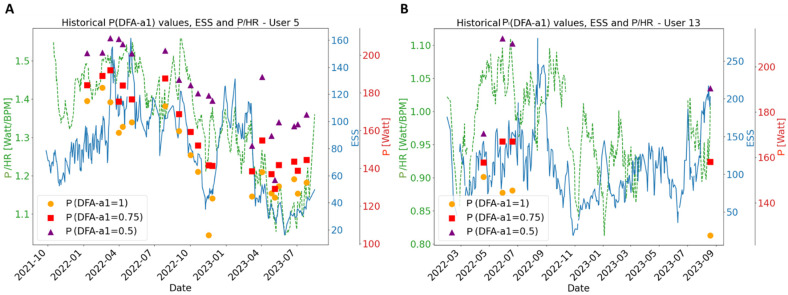
Historical trends of P(DFA-a1) at DFA-a1 levels of 1.0, 0.75, and 0.50 from the workout groups with R^2^ > 0.75 plotted against the calculated External Stress Score (ESS, dimensionless, in blue) and the power-to-HR ratio (P/HR in Watts/BPM, Beats Per Minute, in green) and averaged over a preceding 10-day period. Data are derived from two users (**A**,**B**) out of eleven who provided enough workout groups with reliable fits.

## Data Availability

The data are available upon reasonable request.

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
