# Peer review of "Relationship of Cycling Power and Non-Linear Heart Rate Variability from Everyday Workout Data: Potential for Intensity Zone Estimation and Monitoring"

_sensors, 2024, doi:10.3390/s24144468_

Round 1

Reviewer 1 Report

Comments and Suggestions for Authors

The paper Relationship of cycling power and non linear heart rate  variability The paper titled "Relationship of Cycling Power and Non-linear Heart Rate Variability from Everyday Workout Data: Potential for Intensity Zone Estimation and Monitoring" is well presented, showcasing some novelty and providing interesting insights.

However, I have one concern regarding the DFA data analysis. The paper does not specify how many points the authors used to compute DFA. This detail is crucial because DFA is a fractal method, and the number of points can significantly influence the reliability and accuracy of the results. Providing this information would enhance the reproducibility and robustness of the study's findings.

Overall, the paper has potential, but addressing this issue would strengthen its contribution to the field.

Author Response

Dear editor, dear reviewer, thank you for your comments and time for reviewing our manuscript “Relationship of cycling power and non-linear heart rate variability from everyday workout data: Potential for intensity zone estimation and monitoring“. Below we provide a point-by-point response to each comment.

sensors-3053881

---------------------------------------------------------------------------

Reviewer 1

The paper titled "Relationship of Cycling Power and Non-linear Heart Rate Variability from Everyday Workout Data: Potential for Intensity Zone Estimation and Monitoring" is well presented, showcasing some novelty and providing interesting insights.

However, I have one concern regarding the DFA data analysis. The paper does not specify how many points the authors used to compute DFA. This detail is crucial because DFA is a fractal method, and the number of points can significantly influence the reliability and accuracy of the results. Providing this information would enhance the reproducibility and robustness of the study's findings.

Overall, the paper has potential, but addressing this issue would strengthen its contribution to the field.

ANSWER:

Thank you, we are thankful for the invested time and consideration of our manuscript.

We included this information from L125: „DFA-a1 was calculated with “short term fluctuations” 4 ≤ n ≤ 16 beats. For the DFA-a1 calculation, the window width was set to 120 s, with a recalculation grid interval performed every 5 s. The same window width of 120 s was applied to calculate the cycling power (in Watts) as the mean over this window, every 5 s.“

Reviewer 2 Report

Comments and Suggestions for Authors

Comments to Authors:

Line 177: ‘4’ should be four.

Line 139-140: It may be beneficial for the reader to list the average length of each workout included in the data analysis. This will be beneficial in terms of the ‘why’ behind choosing 20 minutes as the top range of ‘fatigue’. Furthermore, it would be beneficial to defend the reasoning behind choosing the 20-minute top range.

Line 146: Grammar issue; Sentence should read ‘…stopping spinning for a few….’

Line 150-151: Please provide a definition for ‘prolonged’ and ‘frequent periods’.

Line 251: ‘Was’ should be ‘were’.

This is a very novel study. The weaknesses appear to be somewhat challenging, but everything has been properly acknowledged and fully described by the authors.

Overall, while there are major weaknesses and limitations, this study helps provide a foundation from which future research can be conducted to help reveal more specific intensity zone demarcation metrics.

Comments on the Quality of English Language

There are a few minor edits, as detailed in the Comments to Authors

Author Response

Dear editor, dear reviewer, thank you for your comments and time for reviewing our manuscript “Relationship of cycling power and non-linear heart rate variability from everyday workout data: Potential for intensity zone estimation and monitoring“. Below we provide a point-by-point response to each comment.

sensors-3053881

---------------------------------------------------------------------------

Reviewer 2

Line 177: ‘4’ should be four.

ANSWER:

Thank you, we changed it accordingly.

Line 139-140: It may be beneficial for the reader to list the average length of each workout included in the data analysis. This will be beneficial in terms of the ‘why’ behind choosing 20 minutes as the top range of ‘fatigue’. Furthermore, it would be beneficial to defend the reasoning behind choosing the 20-minute top range.

ANSWER:

Thank you, the average length of each workout included was 51 minutes. There is no essential benefit to add this information, since the decision to use the first 20 minutes of each workout as data reference (minute 6-20) is not dependent on that metric. There is no scientific reasoning using the first 20 minutes, its a compromise of data length (having numerous data points depending on heart rate and RR interval length) and assumption about decoupling/fatigue mechanims of external-to-internal load relationship.

Line 146: Grammar issue; Sentence should read ‘…stopping spinning for a few….’

ANSWER:

Thank you, we changed it.

Line 150-151: Please provide a definition for ‘prolonged’ and ‘frequent periods’.

ANSWER:

Thank you, we included this information in L159: „To maintain consistency, workouts with prolonged or frequent periods of no pedaling (exclusion of data sets pedaling <90% of the data set) were excluded and data points with a power of 0 were discarded.“

Line 251: ‘Was’ should be ‘were’.

ANSWER:

Thank you, we changed it accordingly.

This is a very novel study. The weaknesses appear to be somewhat challenging, but everything has been properly acknowledged and fully described by the authors.

Overall, while there are major weaknesses and limitations, this study helps provide a foundation from which future research can be conducted to help reveal more specific intensity zone demarcation metrics.

ANSWER:

Thank you for the invested time and seeing potential in this approach.